# Dynamic Optimization of Lactic Acid Production from Grape Stalk Solid-State Fermentation with *Rhizopus oryzae* Applying a Variable Temperature Profile

María Carla Groff [1,2,*], Sandra Edith Noriega [3], Rocío Mariel Gil [1], Nadia Pantano [2,4] and Gustavo Scaglia [2,4]

1 Instituto de Biotecnología, Facultad de Ingeniería, Universidad Nacional de San Juan (IBT-FI-UNSJ), San Juan 5400, Argentina; rocio.mariel.gil@gmail.com
2 Consejo Nacional de Investigaciones Científicas y Técnicas (CONICET), Buenos Aires 1425, Argentina; npantano@unsj.edu.ar (N.P.); gscaglia@unsj.edu.ar (G.S.)
3 Instituto de Investigaciones en Ciencias Químicas, Facultad de Ciencias Químicas y Tecnológicas, Universidad Católica de Cuyo (IICQ-FCQT-UCCuyo), San Juan 5400, Argentina; sandraneb@yahoo.com
4 Instituto de Ingeniería Química, Facultad de Ingeniería, Universidad Nacional de San Juan (IIQ-FI-UNSJ), San Juan 5400, Argentina
* Correspondence: mcarlagroff@gmail.com; Tel.: +54-9-264-543-6303

**Abstract:** Lactic acid is widely used in the food industry. It can be produced via chemical synthesis or biotechnological pathways by using renewable resources as substrates. The main challenge of sustainable production lies in reaching productivities and yields that allow for their industrial production. In this case, the application of process engineering becomes a crucial tool to improve the performance of bioprocesses. In this work, we performed the solid-state fermentation of grape stalk using *Rhizopus oryzae* NCIM 1299 to obtain lactic acid, employing three different temperatures (22, 35, and 40 °C) and a relative humidity of 50%. The Logistic and First-Order Plus Dead Time models were adjusted for fungal biomass growth, and the Luedeking and Piret with Delay Time model was used for lactic acid production, obtaining higher $R^2$ values in all cases. At 40 °C, it was observed that *Rhizopus oryzae* grew in pellet form, resulting in an increase in lactic acid productivity. In this context, the effect of temperature on the kinetic parameters was evaluated with a polynomial correlation. Finally, using this correlation, a smooth and continuous optimal temperature profile was obtained by a dynamic optimization method, improving the final lactic acid concentration by 53%.

**Keywords:** lactic acid; solid-state fermentation; variable profile temperature; dynamic optimization





## 1. Introduction

Lactic acid (LA) is an organic acid widely used in industrial-scale processes. One of the main fields of application is the food industry, representing 24% of the total demand [1]. It has diverse functional properties, such as lowering the pH of meats to inhibit spoilage bacteria, aiding in the manipulation and molding of chocolates and candies, enhancing chocolate flavor, reducing sucrose inversion in candy production, and regulating the pH of dairy, pickles, baked goods, and beverages, among other functions [2].

In general, LA exists as two pure optical isomers: L(+) lactic acid and D(−) lactic acid, or as a racemic mixture of both [3]. The D(−) lactic acid isomer, in doses exceeding 100 mg·kg body mass$^{-1}$·day$^{-1}$, is harmful to the human body, as the digestive system only possesses the L-lactate dehydrogenase (L-LDH) enzyme, metabolizing L(+) lactic acid. This can lead to acidosis and demineralization [4,5]. Hence, in the food, cosmetic, and pharmaceutical industries, it is essential to use the L(+) lactic acid isomer, produced with varying degrees of purity (ranging from 50 to 90%), complying with relevant legislation [6]. On an industrial scale, LA can be produced via chemical synthesis using petroleum precursor raw materials (for example, acetaldehyde and hydrogen cyanide yield lactonitrile, followed by

its acid hydrolysis), resulting in a racemic mixture of LA. Conversely, through biotechnological pathways, by utilizing microorganisms that ferment carbohydrates, optically pure L(+) or D(−) lactic acid can be obtained [7].

Currently, 90% of the global LA production occurs through sugar fermentation due to lower raw material costs, employing low process temperatures, being more environmentally friendly, and yielding high-purity LA [8]. However, several aspects still require improvement, such as avoiding raw materials competing with food production, enhancing bioprocess performance, and reducing raw material costs overall [5]. Hence, a promising alternative arises in using agroindustrial residues as fermentation substrates to obtain LA. TripleW, a company specializing in the production of lactic acid from food waste, is now operational in Israel. The process involves the hydrolysis of organic waste, followed by the fermentation and purification of the lactic acid. This versatile lactic acid can be utilized in the production of biomaterials (specifically those based on polylactic acid), as well as in the generation of green energy and organic fertilizers [9].

Regarding the use of renewable resources for fermentative LA production, substrates can be liquid (wastes from the dairy industry, juices, and beverages, among others) or solid starch (sugarcane, potato, corn, rice, and wheat residues), lignocellulosic (wood, tree and fruit pruning residues), and rich in simple sugars (fruit processing residues) [2,5]. In this context, residues generated in the wine industry hold significant potential as raw materials for sustainable bioprocesses. Grape stalk (GS), a residue generated in the initial grape processing for winemaking, has been studied by our research group as a solid-state fermentation (SSF) substrate inoculated with *Rhizopus oryzae* (*R. oryzae*) for LA production [10]. The primary challenge in using GS as a substrate for SSF on a large scale is achieving productivity and yields comparable to those when using pure raw materials. This is where process engineering becomes crucial and highly interesting for application in bioprocesses projected for larger scales.

The mathematical modeling of bioprocesses serves as the initial step in implementing process engineering in biotechnology. It involves developing a mathematical model describing the most important mechanisms of the biological system (cell growth kinetics, product formation, and substrate consumption) and their interaction within the bioprocess parameters [11]. In the case of SSF, models are applied to microbial growth kinetics, such as empirical models (the Logistic model and two-phase model) and pathway-based models, with the Luedeking–Piret model being the most used for metabolite production [12]. In this regard, our research group has applied the First-Order Plus Dead Time model (FOPDT) to cell growth due to its mathematical simplicity and widespread usage and recognition among process engineers [10,13,14]. In addition, we previously proposed a modification of the Luedeking and Piret model [13], in which we introduced a time delay parameter ($t_d$); this parameter accounts for the lag time that exists between the production of biomass and LA. Therefore, the Luedeking and Piret with Delay Time model (LPwDT) will be used to fit the experimental data of LA production.

Regarding the dependency of the kinetic parameters of SSF on temperature, some works were found relating to the relationship of kinetic parameters with temperature. In [15], the Rosso, Ratkowsky, Zwietering, Esener, and Arrhenius models were employed to correlate the dependency of kinetic parameters for cellulase production by *Aspergillus niger* (*A. niger*) with temperature. The authors found that only the maximum biomass concentration ($X_{max}$) and maximum growth rate ($\mu_{max}$) parameters had an adequate fit with the Rosso and Ratkowsky models. Furthermore, in [15], SSF was conducted using *A. niger* to produce enzymes, revealing that the Rosso model exhibited a high level of fitting to relate $\mu_{max}$, $X_{max}$, and the product formation constant associated with growth ($\alpha$ or $Y_{p/x}$) with temperature. But with respect to fungal SSF for obtaining LA, no studies were found that relate kinetic parameters to temperature, so the present study is an important contribution in this area.

Once the parameters of the bioprocess mathematical model have been identified, the optimization stage follows, defining the operational variables (temperature, flow rate,

and pH, among others) that allow for the best bioprocess performance to be achieved (maximizing metabolite production and minimizing environmental impact, among others). Traditional optimization involves generating a response surface from a set of experiments to find the optimal operating condition to fulfill a specific objective (e.g., maximizing metabolite quantity) [16]. However, in industrial biotechnology, parameters (such as temperature) are often maintained at a constant optimum value, and little exploration has been carried out on using a variable temperature profile to maximize a bioproduct's yield [17]. In this regard, our research group developed a dynamic optimization technique based on Orthogonal Polynomials and Fourier Series [18,19] specifically designed for bioprocesses (biodiesel, biogas xylitol, and recombinant protein production), enabling the attainment of smooth profiles for the studied physicochemical variable (continuous and differentiable, avoiding abrupt changes that negatively affect microorganisms). However, dynamic optimization studies using variable profiles have not been applied to LA production by SSF.

In this research, SSF trials were performed utilizing GS as a bioresource and *R. oryzae* NCIM 1299 as the microorganism responsible for LA production. The experiments were conducted at three different constant temperatures (22, 35, and 40 °C) while maintaining the relative humidity at 50%. Then, the Logistic and FOPDT models were used to describe the biomass generation, and the LPwDT model was used to describe the LA production at the three constant temperatures. A parametric identification based on evolutive algorithms was proposed to adjust the models to the experimental data. Once the kinetic parameters were identified, a second-degree polynomial correlation was fitted to evaluate the effect of temperature on the kinetic parameters. Finally, an optimal temperature profile was determined by a dynamic optimization method (using Orthogonal Polynomial and Fourier Series methods) to improve the final LA concentration.

## 2. Materials and Methods

### 2.1. Solid-State Fermentation

Solid-state fermentation was carried out as described in [10], for 120 h in Petri dishes with membrane filter culture. This cultivation system was assembled as follows: at the bottom of the Petri dishes, 5 mL of the culture medium formulated with grape stalk and agar (Agar + GS = 0.05 g GS/mL of medium) was dispersed. Above the Agar + GS, a 0.2 μm nylon membrane (sterilized, dry, and tared) was placed. Once this was completed, each Petri dish was inoculated with a suspension of *R. oryzae* NCIM 1299 (from Centro de Referencia de Micología, Facultad de Ciencias Bioquímicas y Farmacéuticas, Universidad Nacional de Rosario, Rosario, Argentina)o at a known concentration ($1.37 \times 10^8$ spores/mL). In order to obtain the relationship of the kinetic parameters with temperature, twenty-four Petri dishes were placed in an incubator with temperature and relative humidity control (SEMEDIC, L-291PH). The samples were collected in duplicate every 12 h. The following procedure was applied to each sample: the filter over which the *R. oryzae* had grown was removed to determine the fungal dry biomass through gravimetry (at 80 ± 5 °C for 24 h). This dry biomass was subjected to acid hydrolysis with $H_2SO_4$ to break down fungal structures, enabling the quantification of the generated LA using a spectrophotometric technique (quantification at 390 nm of the ferric lactate produced by the reaction between the lactate ion and the ferric ion from ferric chloride) [20].

The mean values and standard deviation (SD) from duplicate experiments were used to generate growth of biomass and LA production graphs for each time point.

Constant temperatures of 22, 35, and 40 °C at 50% RH were chosen in order to cover a wide range of temperature with the next suppositions:

(a) When using a small-scale experimental growth chamber (membrane filter culture system), heat transfer phenomena are not limiting. Therefore, it can be assumed that the solid substrate is at the temperature and ambient relative humidity of the incubator chamber air.

(b) Petri dishes were incubated at different constant temperatures (maintaining constant relative humidity). The ranges of temperature were chosen to encompass optimal

parameters with the minimum number of trials. These conditions remained constant in each culture throughout the growth cycle.

(c)     All Petri dishes were inoculated with the same spore suspension at the same concentration.

(d)     In SSF, pH is a very difficult parameter to measure and control. Additionally, *R. oryzae* is a strain capable of growing in a range of pH from 4 to 9 [21]. So, the pH was neither measured nor controlled in the experiments.

### 2.2. Mathematical Bioprocess Model

The biomass and LA experimental results were plotted as a function of time, and the mathematical models were fitted using Matlab R2015a software. The Logistic model (Equation (1)) and FOPDT model (Equations (2) and (3)) were adjusted for fungal biomass growth [10,13]:

$$\frac{dX}{dt} = \mu_{max}\left(1 - \frac{X}{X_{max}}\right)X, \ X(0) = X_0 \tag{1}$$

$$\frac{dX}{dt} + \frac{X}{t_p} = \frac{1}{t_p}\cdot X_{max}(t - t_0), \ X(0) = X_0 \tag{2}$$

where the function $X_{max}$ is defined by

$$X_{max}(t - t_0) = \begin{cases} X_{max} & for \quad t \geq t_0 \\ 0 & for \quad t < t_0 \end{cases} \tag{3}$$

where $X(t)$: the fungal biomass concentration obtained for a specific amount of time [g dry biomass/g dry grape stalk, g DB/g GS]; $\mu_{max}$: the maximum specific growth rate [1/h]; $X_{max}$: the maximum biomass concentration [g DB/g GS]; $X_0$: the *R. oryzae* inoculum [g DB/g GS], $t_0$: the time range in which there is no fungal biomass growth (latency phase) [h]; $t_p$: the parameter of the bioprocess that provides information on the speed of growth up to $X_{max}$ [h]; and $t$: time [h].

By applying the FOPDT model to fungal growth at the different working temperatures, the parameters $t_p$ and $t_0$ were obtained. This model has not been described or developed in other microbial kinetic works. In this context, $t_p$ is an inverse parameter to $\mu_{max}$, since the $t_p$ parameter is described in units of time, while $\mu_{max}$ is the inverse of time.

LA is a primary metabolite, so the LA generation speed has a direct relationship with the *R. oryzae* growth rate ($dX/dt$), with the LA yield ($Y_{p/x}$) being different from 0 and the coefficient for LA production related to maintenance metabolism ($m_p$) being equal to 0. For instance, the Luedeking and Piret with Delay Time model (LPwDT) [13] was used (Equations (4) and (5)):

$$\frac{dP}{dt} = Y_{p/x}\frac{dX(t - t_d)}{dt} \ for \ P(0) = P_0 = 0 \tag{4}$$

where the function $\frac{dX(t - t_d)}{dt}$ is defined by

$$\frac{dX(t - t_d)}{dt} = \begin{cases} \frac{dX}{dt} \ for \ t \geq t_d \\ 0 \ for \ t < t_d \end{cases} \tag{5}$$

where $dP/dt$: the LA production rate [g LA/g GS·h]; $Y_{p/x}$: the LA yield [g LA/g DB]; $P_0$: the initial LA concentration [g LA/g GS]; and $t_d$: the time difference between when LA production and biomass generation starts [h].

In order to use the Logistic model combined with the LPwDT model (L-LPwDT model), Equation (1) was substituted into Equation (4). To apply the FOPDT model combined with the LPwDT model (FOPDTL-LPwDT model), Equation (2) was substituted into Equation (4).

2.2.1. Parametric Identification

The parametric identification proposed for each of the experiments at different temperatures consists of a hybrid methodology originally designed in [22], in which the technique combines the Monte Carlo and Genetic Algorithm methods. The use of the Monte Carlo method accelerates the convergence of the Genetic Algorithm, refining the results and enabling the identification of the best set of parameters with minimal error. The following parameters were identified: $X_{max}$, $\mu_{max}$, $t_p$, $t_0$, $Y_{p/x}$, and $t_d$. The algorithm was designed by finding the maximum value of R$^2$ for each model:

$$R^2 = \left(1 - \frac{SS_{RES}}{SS_{TOT}}\right)100 = \left(1 - \frac{\sum_i (y_i - \hat{y}_i)^2}{\sum_i \left(y_i - \overline{\overline{y}}_i\right)^2}\right)100 \tag{6}$$

where $SS_{RES}$ is the sum of squares of the distance between the real point ($y_i$) and the predicted point in the model ($\hat{y}_i$), and $SS_{TOT}$ is the sum of squares of the distance between the real point ($y_i$) and the mean of all points of the mean line ($\overline{y}$).

The software Matlab R2015a was used to test the fitting mathematical models for kinetic growth and LA production.

2.2.2. Effect of Temperature Polynomial Relationship on the Kinetic Parameters of *R. oryzae* Growth and Lactic Acid Production

In this study, a polynomial adjustment of the experimental kinetic results was carried out since it is easier to manipulate mathematically. This methodology has already been applied in liquid fermentation [23,24]. Subsequently, the parameters $\mu_{max}$, $X_{max}$, $t_p$, $t_0$, and $Y_{p/x}$ were plotted as a function of temperature, using the curve-fitting application in Matlab R2015a, selecting the second-grade polynomial model:

$$f(x) = p_1 x^2 + p_2 x + p_3 \tag{7}$$

where $p_1$, $p_2$, and $p_3$ are the adjustment parameters, and $x$ is the temperature (T). The polynomials obtained were used as input information to obtain the variable temperature profile that maximizes LA production, through the implementation of the dynamic optimization technique, based on Orthogonal Polynomials and Fourier Series.

*2.3. Dynamic Optimization of LA Production*

The temperature-dependent mathematical models obtained in Section 2.2.2 were used to determine the optimal temperature profile, T(t), that maximizes the concentration of LA at the final time (120 h). For this, a strategy based on Orthonormal Polynomials and Fourier Series was proposed for the control vector parameterization, combined with the hybrid technique based on evolutive algorithms for the parameter optimization described in Section 2.2.1. The dynamic optimization methodology was originally proposed by [19], and it has great advantages over conventional techniques. First, the resulting profiles are continuous and differentiable and do not have abrupt changes in the control variables. This is very beneficial when working with bioprocesses since stress and cell death are avoided. Second, and no less important, the number of parameters required for optimization is minimal. The optimization procedure was carried out in the simulation environment of Matlab R2015a.

The statement of the optimal control problem (OCP) was formulated once the mathematical model was completed, and it is presented in Section 3.3.

**3. Results and Discussion**

*3.1. SSF and Mathematical Bioprocess Model*

As is well known, temperature directly affects the kinetics of microbial growth. Figure 1 presents the experimental results (averages with their respective standard deviations) of the *R. oryzae* growth and LA kinetics in the experiments at the different temperatures studied.

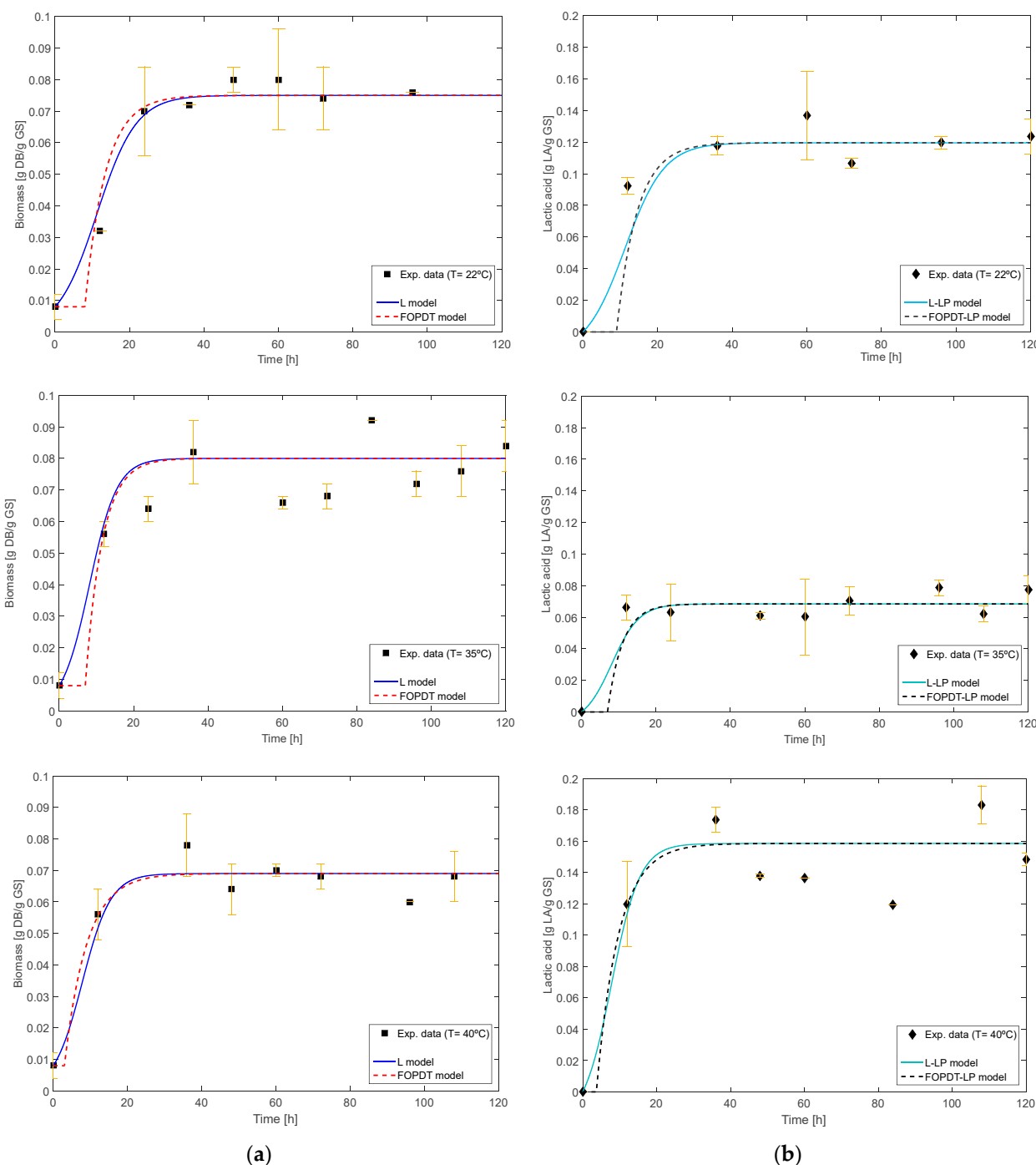

**Figure 1.** Comparison between the experimental data and mathematical models for (**a**) fungal cell growth kinetics and (**b**) lactic acid production kinetics.

Regarding the production of fungal biomass by *R. oryzae*, it could be observed that there were no significant differences in the curves at the different temperatures studied. This could be attributed to the high adaptation of *R. oryzae* to grow within a broad temperature range from 7 to 45 °C [21], resulting in minimal discrepancy in the biomass generated at the working temperatures. Based on these results, the Logistic and FOPDT models were fitted to the growth, and the kinetic parameters are compiled in Table 1.

**Table 1.** Mathematical models' parameters of fungal and lactic acid kinetics at different constant temperatures.

| Ambiental Conditions | | | *R. oryzae* Growth | | | | | | Lactic Acid Production | | | | |
| --- | --- | --- | --- | --- | --- | --- | --- | --- | --- | --- | --- | --- | --- |
| | | | Logistic Model | | | FOPDT Model | | | L-LPwDT Model | | FOPDT-LPwDT Model | | |
| RH | T | $X_0$ | $X_{max}$ | $\mu_{max}$ | $R^2$ | $t_p$ | $t_0$ | $R^2$ | $Y_{p/x}$ | $R^2$ | $Y_{p/x}$ | $t_d$ | $R^2$ |
| | 22 | | $0.075 \pm 0.002$ | $0.189 \pm 0.006$ | 96.15 | $5.50 \pm 0.14$ | $9.00 \pm 0.27$ | 97.36 | $1.785 \pm 0.053$ | 82.45 | $1.785 \pm 0.053$ | 0 | 79.23 |
| 50 | 35 | $0.008 \pm 0.002$ | $0.080 \pm 0.002$ | $0.270 \pm 0.008$ | 82.77 | $4.50 \pm 0.12$ | $7.00 \pm 0.20$ | 83.55 | $0.95 \pm 0.03$ | 83.76 | $0.95 \pm 0.03$ | 0 | 83.92 |
| | 40 | | $0.069 \pm 0.002$ | $0.260 \pm 0.008$ | 89.38 | $6.00 \pm 0.17$ | $3.00 \pm 0.08$ | 91.60 | $2.60 \pm 0.13$ | 84.33 | $2.60 \pm 0.13$ | $0.70 \pm 0.02$ | 84.40 |

After fitting the Logistic and FOPDT models to the experimental growth points of *R. oryzae* at the different temperatures studied, it could be observed (see Table 1) that, in all three cases, the FOPDT model exhibited higher $R^2$ values than the Logistic model. This suggests that it is a very good model proposal for the growth of *R. oryzae* in SSF, as previously concluded in [10]. In the case of the LPwDT model fit, Table 1 shows that there were no significant differences in the $R^2$ value when applying the Logistic model (L-LPwDT model) or the FOPDT model (FOPDT-LPwDT), with a slight improvement in the $R^2$ value for the Logistic model. Although there were no significant differences in the fit of both models, an advantage was observed when applying the FOPDT model. This advantage lies in the ability to modify the value of $t_0$ while maintaining the same value of $t_p$ as that for fungal growth. This observation might indicate the presence of a time delay ($t_d$) between biomass and LA production, a phenomenon that was examined and described in [13]. In the present study, no time delay was found between LA production and biomass generation at temperatures of 22 °C and 35 °C (i.e., $t_d = 0$). But when working at 40 °C, a time delay was observed ($t_d = 0.7$ h). This could be explained by the change in the fungal morphology to pellet form observed at 40 °C (this point is explained in the following paragraphs).

At 35 °C, a higher production of fungal biomass was achieved than at 22 °C and 40 °C, although the difference between them was not significantly notable. In [25], the authors worked with *A. niger* in SSF using cassava pulp and soybeans as solid substrates, observing a significant difference in biomass generated at different temperatures while maintaining a constant substrate humidity of 50% (they reported the following: 0.130 g DB/g dry solid substrate at 20 °C, 0.266 g DB/g DSS at 30 °C, 0.188 g DB/g DSS at 40 °C, and 0.106 g DB/g DSS at 50 °C). Similarly, in [26], a significant variation in $X_{max}$ content concerning temperature was observed: 78.3 g DB/kg DSS (20 °C), 102.85 g DB/kg DSS (25 °C), 127.36 g DB/kg DSS (30 °C), 114.70 g DB/kg DSS (35 °C), and 53.19 g DB/kg DSS (40 °C). However, in [27], the authors used *A. niger* in a membrane growth system with wheat bran, and they concluded that the $X_{max}$ content is independent of temperature, as they did not observe a significant difference but found it to be dependent on the solid substrate's humidity. Based on this, it can be stated that our results are consistent with those of other SSF studies, even though the kinetic growth information of *R. oryzae* in SSF has not been found.

Temperature not only affects microbial growth kinetics but also has a direct impact on the production of microbial metabolites, such as LA in this case. An increased production of LA was observed at 40 °C, followed by fermentation at 22 °C and then at 35 °C (see Figure 1). This trend was also reflected in the $Y_{p/x}$ values adjusted with the LPwDT model (see Table 1). In order to explain this situation, we observed the results obtained in [28], in which the authors conducted a study using starchy potato effluents in liquid fermentation to produce LA with *R. oryzae* and *R. arrhizus*. They evaluated the effects of pH, the supplementation of $CaCO_3$, temperature, and carbonaceous substrates on biomass and LA generation. Concerning temperature influence, they worked at 22 °C, 30 °C, 35 °C, and 40 °C, finding favorable LA production within the temperature range of 30 to 40 °C for *R. oryzae*. They also observed that *R. oryzae* exhibits a relatively slower rate (compared to *R. arrhizus*) in hydrolyzing starch into glucose, potentially leading to lower substrate (glucose) inhibition on LA production. However, regarding LA yield, they obtained a very low value of 0.40 g/g (compared to *R. arrhizus*, which presented a value of 0.80 g/g). Another crucial result (although not demonstrated in their study) was that *R. oryzae* had a higher biomass yield than *R. arrhizus*. Based on these outcomes, the authors concluded that there might be a competition for carbon sources between biomass formation and LA production in the fermentation process. A similar phenomenon might occur in the SSF in the present study, where, at the optimal temperature for biomass generation, there was a noticeable decrease in LA production. This could potentially be due to the high enzymatic activity of *R. oryzae*, leading to an increase in glucose and/or xylose content, consequently inhibiting LA production. Conversely, at extreme temperatures where biomass generation and enzymatic activity are lower, LA production might be favored.

Another aspect to consider in LA production is the accumulation of fermentative byproducts, such as ethanol and fumaric acid, since the metabolism of *R. oryzae* is heterofermentative [1]. Although the produced quantities of fumaric acid and ethanol are much smaller than the produced quantity of LA (which is the primary metabolite of *R. oryzae*) [29–31], these byproducts could potentially influence the low LA production at 35 °C. However, the presence of byproducts in the present study could not be demonstrated, as only LA was quantified, thus suggesting a potential avenue for future investigation. To address this, a spectrophotometric technique for quantifying fumaric acid was proposed in [32], similar to the quantification of LA applied in the present work, which could be implemented in future studies.

### 3.2. Effect of Temperature Polynomial Relationship on the Kinetic Parameters of R. oryzae Growth and Lactic Acid Production

Regarding the fitting of equations relating $X_{max}$ to temperature, as shown in Table 2 and Figure 2a, a better temperature of 29.92 °C was found, with an $X_{max}$ of 0.0838 g DB/g GS. With respect to $\mu_{max}$, a maximum of 0.27 h$^{-1}$ was observed at 35 °C (see Figure 2b). Although not at the same temperature as the optimal $X_{max}$ value, they are very close to each other. In [25], $X_{max}$ and $\mu_{max}$ exhibit the same temperature optimum value, while in [27], they differ from each other; both worked with *A. niger*. This situation could depend on the specific metabolism of the fungus in the development conditions under study, being that, under certain conditions, it could have optimal growth rates at a certain temperature, while the maximum biomass content could be found under another condition. It could be concluded that the best values of $\mu_{max}$ and $X_{max}$ were obtained in a temperature range of 30 to 35 °C. It is important to highlight that the aim of this work was to maximize LA production. Thus, even if a low value of $X_{max}$ was obtained at the optimum temperature for LA production (when dynamic optimization applies), it would be beneficial for SSF due to the problems generated by the mycelium of the fungus during the mass and caloric transfers.

**Table 2.** Polynomial relation between model parameters and temperature.

| Parameters | Fitted Polynomial | $R^2$ |
|:---:|:---:|:---:|
| $X_{max}$ | $X_{max}$ (T) $= -0.00014$ T$^2$ + $0.00857$ T $- 0.04403$ | 100 |
| $\mu_{max}$ | $\mu_{max}$ (T) $= -0.00046$ T$^2$ + $0.0323$ T $- 0.3002$ | 100 |
| $t_p$ | $t_p$ (T) $= 0.021$ T$^2$ $- 1.271$ T + $23.32$ | 100 |
| $t_0$ | $t_0$ (T) $= -0.0359$ T$^2$ + $1.892$ T $- 15.26$ | 81.76 |
| $Y_{p/x}$ | $Y_{p/x}$ (T) $= 0.0219$ T$^2$ $- 1.313$ T + $20.06$ | 100 |

By applying the FOPDT model to fungal growth at the different working temperatures, the parameters $t_p$ and $t_0$ were obtained. Therefore, in this case, a minimum $t_p$ was sought at which the microorganism is able to reach 63.2% of $X_{max}$ ($t_p$ definition). The best $t_p$ value (4.04 h) was found at a temperature of 30.53 °C (see Figure 2c). Regarding $t_0$, which could reflect the adaptation time of the microorganism, it was observed (see Figure 2d) that, at low temperatures, it had a slight curvature that made it remain at high values until after a certain temperature, where it decreased abruptly; i.e., as the temperature increased, smaller values of $t_0$ were obtained. This relationship is correct, since an increase in temperature accelerates the metabolic and enzymatic processes of microorganisms, which makes the adaptation time shorter.

Concerning the effect of temperature on the $Y_{p/x}$ parameter, Figure 2e shows an increase in the value of $Y_{p/x}$ as the temperature reaches 40 °C. Different situations have been observed in other fermentation studies. For instance, in [33], liquid fermentation was carried out using *Lactobacillus bulgaricus* to obtain LA, and it was observed that the $Y_{p/x}$ value remained constant (0.7 g LA/g DB) despite the change in temperature. In another study [34], *Brettanomyces bruxellensis* was employed to obtain ethanol and acetic acid in liquid fermentation, and the value of $\alpha$ ($Y_{p/x}$) remained constant with temperature

changes, while the value of $\beta$ ($m_p$) varied with the temperature changes. In contrast, in [26], cellulases were obtained with *A. niger* in SSF, and both parameters ($Y_{p/x}$ and $m_p$) varied with temperature. A similar observation was made in [25], where enzymes were obtained with *A. niger* in SSF, but in this case, the value of $\alpha$ ($Y_{p/x}$) varied more than the value of $\beta$ ($m_p$). In both SSF studies, there was variation in the $Y_{p/x}$ value concerning temperature, resembling the findings in the present study. Therefore, our findings align with the results observed in other SSF studies.

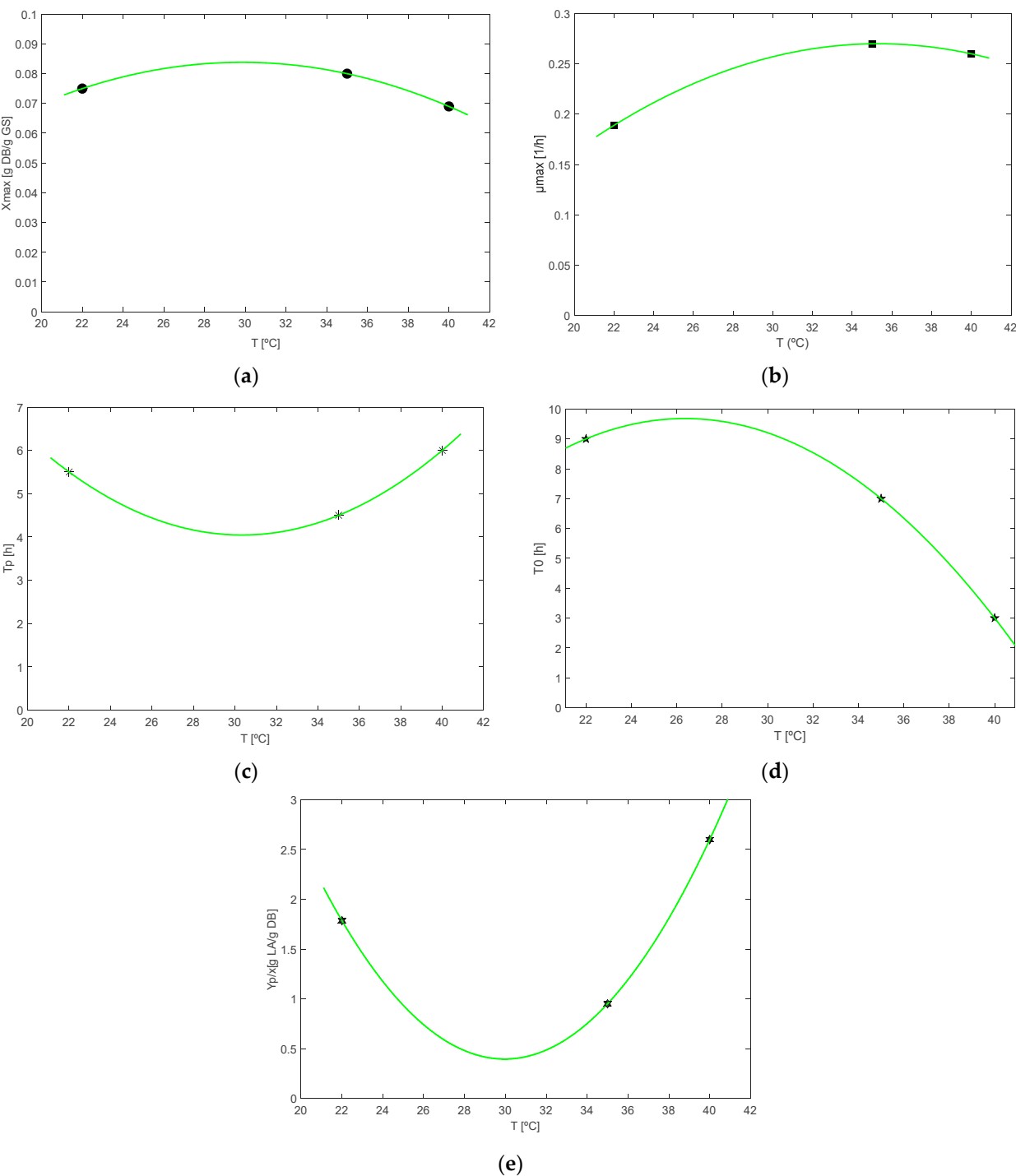

**Figure 2.** Relationship between the model parameters and temperature. (**a**): $X_{max}$ vs. T, (**b**) $\mu_{max}$ vs. T, (**c**) $t_p$ vs. T, (**d**) $t_0$ vs. T, and (**e**) $Y_{p/x}$ vs. T.

### 3.3. Effect of Temperature on the Growth Morphology of R. oryzae NCIM 1299

An important observation derived from cultivating *R. oryzae* at different temperatures was the typical mycelial growth pattern observed at 22 and 35 °C. However, when subjected to 40 °C, a change in the morphology of *R. oryzae* NCIM 1299 was detected, as shown in Figure 3.

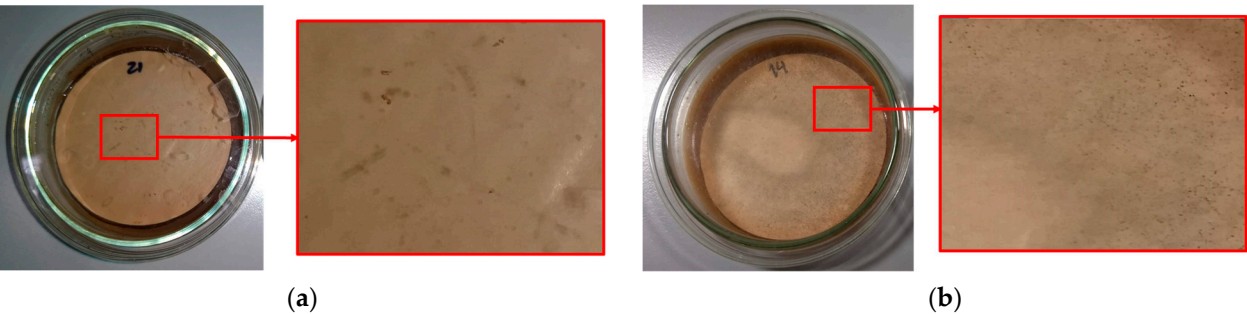

(**a**)  (**b**)

**Figure 3.** Macromorphology of *R. oryzae* NCIM 1299 growth on Agar + GS: (**a**) pellet form (at 40 °C and 50% RH) and (**b**) mycelial form (at 35 °C and 50% RH).

Figure 3a shows the growth of *R. oryzae* NCIM 1299 at 48 h, 40 °C, and 50% RH, showing small pellets distributed on the nylon membrane (seen under zoom). These formations, or pellets, are distributed across the membrane, suggesting that the extreme temperature induced a change in the typical mycelial morphology of the fungus (as shown in Figure 3b).

Fungi are microorganisms widely used at industrial scale to obtain various bioproducts. One of the challenges faced during their development is the ability of these microorganisms to grow as dispersed mycelia or as granules or pellets, influenced by factors such as spore concentration, spore viability, pH, temperature, dissolved oxygen concentration, and mechanical stress [35]. At the industrial level, submerged fermentation is most commonly used, and pellet formation offers advantages, including 1. avoiding the 'slimy' growth observed in some fungi; 2. improving aeration efficiency; 3. reducing medium viscosity, enhancing mass transfer conditions; and 4. simplifying mycelium separation after fermentation [36]. All these benefits could be analogously applied to SSF.

The ways in which fungi develop as pellets can be either coagulative or non-coagulative, and these processes are described in [37]. It has been determined that, when the fungus *R. oryzae* forms as a pellet, it does so in a non-coagulative way, in which the spores germinate before the formation of the pellet so that a single spore can form a granule. Also, in [38], the authors correlated the type of pellets formed with the composition of the fungal cell wall. They concluded that the primary cell wall polysaccharide α-1,3-glucan and the extracellular polysaccharide galactosaminogalactan (GAG) contribute to hyphal aggregation in *Aspergillus oryzae*. In contrast, strains with a low amount of or lacking α-1,3-glucan and GAG, such as *R. oryzae*, exhibit hyphae that aggregate loosely (non-coagulative) or disperse in a liquid culture.

An advantage to consider is that the pelleted growth form has been shown to increase the productivity of fungal metabolites. For example, in [39], the authors demonstrated an increase of about 100% in the concentration of LA generated by *R. oryzae* in pellet form with respect to mycelial growth. Also, in [32], it was found that the pellet form generated the highest amount of fumaric acid when using *R. oryzae* 1526 with brewery wastewater as a fermentative medium.

The change in the growth morphology of *R. oryzae* has been investigated for over two decades, as reflected in [36], in which the authors studied various factors affecting growth morphology in submerged fermentation, such as pH and the trace metal content in the culture medium. However, it is important to note that no studies were found where the fungus *R. oryzae* developed in pellet form in SSF.

Based on the *R. oryzae* morphological change observed, it was decided to proceed with a microscopic observation of the mycelial structure that developed at 35 °C and the pellet structure that developed at 40 °C, as shown in Figure 4 (the lactophenol cotton blue staining technique was applied [40], and the fungal samples were imprinted with transparent adhesive tape to allow for a more detailed examination).

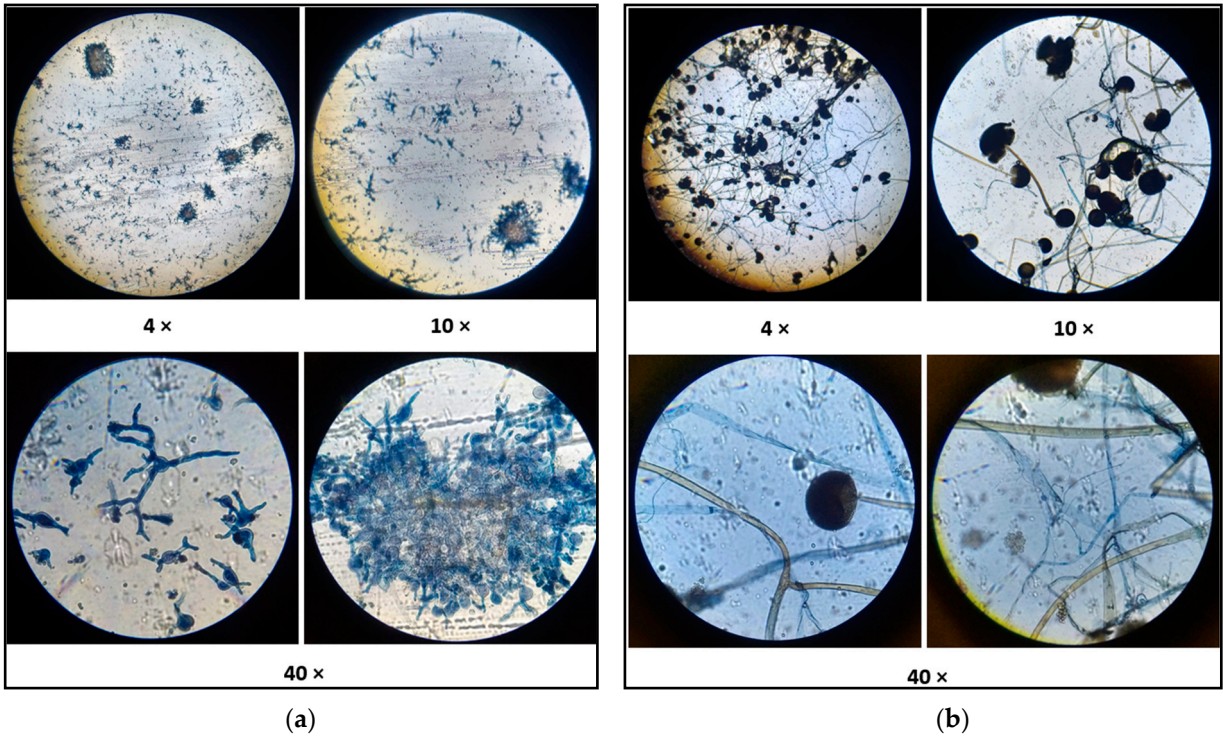

**Figure 4.** Micromorphology of *R. oryzae* NCIM 1299 growth on Agar + GS: (**a**) pellet form (at 40 °C and 50% RH) and (**b**) mycelial form (at 35 °C and 50% RH).

Figure 4a shows that the pellets formed by *R. oryzae* NCIM 1299 are loose aggregates, not very compact, which is supported by what was explained above, while Figure 4b shows the typical morphological structure of *R. oryzae*, with the presence of sporangiophores with globose and multispored black sporangia.

In [41], a similar behavior of *R. oryzae* in response to temperature was observed, showing filamentous growth at lower temperatures between 4 and 34 °C and a yeast-like growth form in a range of 40 to 48 °C. Additionally, the authors demonstrated that this growth type does not revert to filamentous development upon reducing the temperature, possibly due to an irreversible change in the protein nature within the fungal structure.

As shown in [42], it is currently feasible to control and design the pellet morphology based on the need to produce specific metabolites by modifying various external factors. This field of study is called morphological engineering. For example, in [43], the authors were able to form compact and uniform granules of *R. oryzae* by adding triethanolamine to the culture medium, which resulted in a 293% increase in malic acid production and a 177% increase in fumaric acid production. In our work, when mycelial growth occurred (22 and 35 °C), no $t_d$ was observed, and pellet growth occurred at 40 °C, $t_d = 0.7$ h. It is plausible that *R. oryzae* requires additional time to adapt to pellet growth conditions before initiating LA production. Additionally, the $Y_{p/x}$ value at 40 °C was higher than at 22 °C and 35 °C, further supporting the theory that the pellet form is more favorable for increasing fungal metabolite productivities.

Therefore, since most of the studies on fungal morphological engineering have been developed in the field of submerged fermentation, it would be very interesting to consider future work studying the factors that enhance the pellet growth of *R. oryzae* in SSF.

*3.4. Optimal Control Problem Statement*

The statement of the optimal control problem (OCP) consists of finding the best operation temperature profile T(t) that maximizes the performance index, *J*, which is the LA concentration at the final reaction time ($t_f$) of 120 h:

$$J = \left[ LA\left(t_f\right) \right] \tag{8}$$

This objective function is subject to the completed mathematical model (described in Equations (1)–(5), Tables 1 and 2) and the following equality and inequality constraints:

$$\mu_{max} > 0 \; 1/h \tag{9}$$

$$t_p > 0 \; h \tag{10}$$

$$t_0 \geq 0 \; h \tag{11}$$

$$Y_{p/x} \neq 0 \; g \; LA/g \; DB \tag{12}$$

$$t_d \geq 0 \; h \tag{13}$$

$$RH = 50 \; \% \tag{14}$$

$$22 \, °C \; \leq \; T \; \leq 40 \, °C \tag{15}$$

The initial conditions of the system are

$$X_0 = 0.008 \; g \; DB/g \; GS \tag{16}$$

$$P_0 = 0 \; g \; LA/g \; GS \tag{17}$$

The optimization results are shown in Figure 5. The constant and variable temperature profiles are plotted in Figure 5a. In Section 3.2, it was found that the best temperature for obtaining a higher $X_{max}$ content was different from the best temperature that showed the highest LA concentration, being 29.92 °C and 40 °C, respectively. This discrepancy implies that selecting the best constant temperature may not be the optimal choice. However, as demonstrated in Figure 5b,c, employing a variable temperature profile led to increased $X_{max}$ and LA concentrations, with respective percentage increases of 15% and 53% at 120 h.

In the case of biomass concentration, an increase resulting from the use of the variable temperature profile can be seen from 40 h, reaching 0.07922 g DB/g GS at 120 h. In contrast, concerning LA production with the variable temperature profile, an improvement in yield, compared to that obtained at 40 °C, is clearly notable from the beginning of LA production, being already 27% higher at 40 h. Therefore, an increase in the LA yield is observed from the early stages of the bioprocess reaction. This suggests that a variable temperature control strategy could be more attractive for practical fermentation, since shorter process times would result in a higher LA concentration.

In addition, one of the difficulties of SSF when applied on a large scale is removing the heat of the reaction from the solid substrate due to its low conductivity and the fact that the agitation of the bed breaks the structure of the fungal mycelium [44,45]. Therefore, implementing a profile that gradually decreases over time facilitates better removal of the heat generated during the SSF bioprocess.

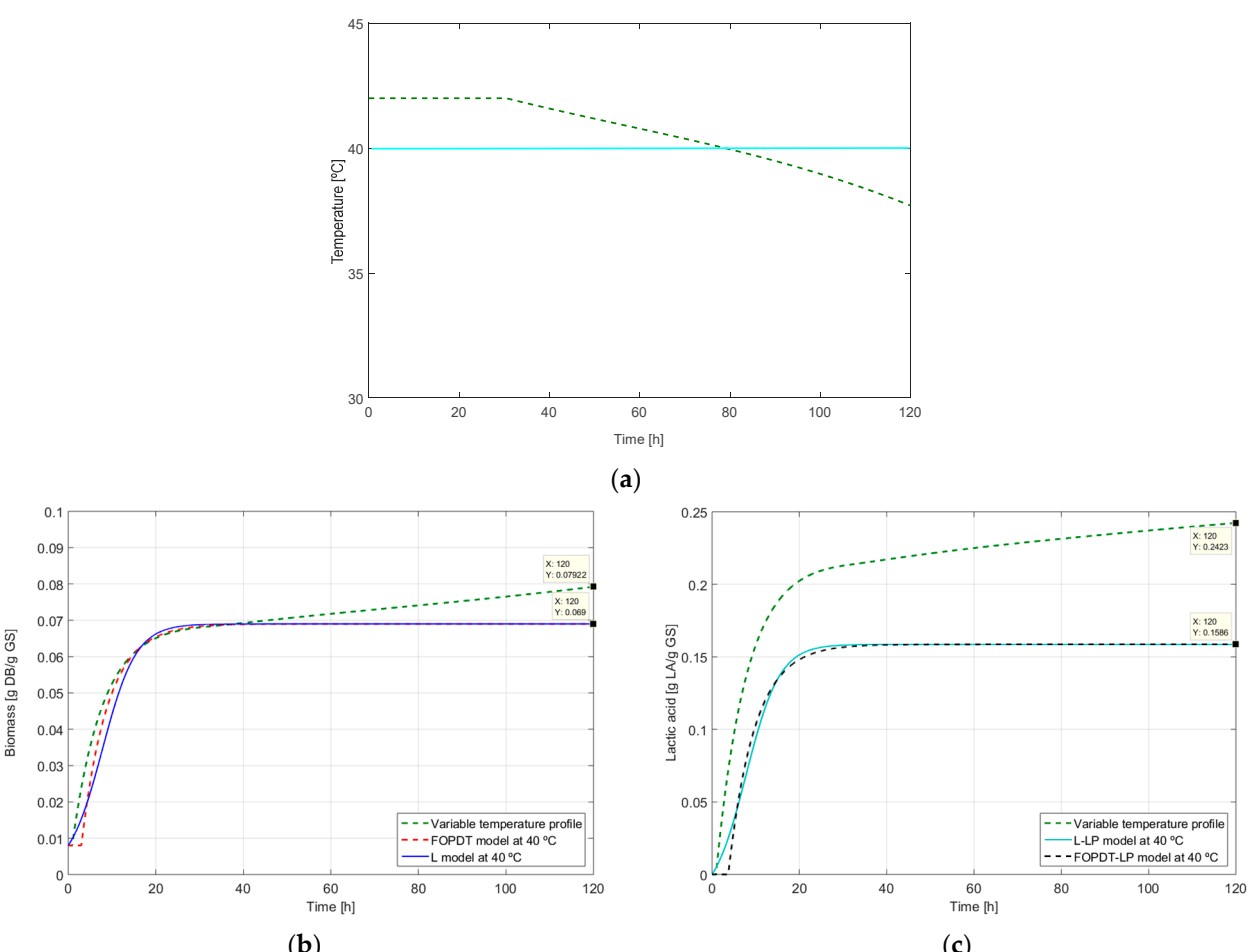

**Figure 5.** Comparison of (**a**) temperature, (**b**) biomass, and (**c**) lactic acid obtained when applying a constant temperature of 40 °C and a variable temperature profile.

No SSF studies that implement temperature control strategies with a variable temperature profile were found, but liquid fermentation studies were found. In [23], a 13.94% increase in triterpene yield was achieved using *Ganoderma lucidum* with a stepped temperature profile from 32 to 29 °C. This profile consisted of four segments maintaining constant temperatures (at 32, 31, 30, and 29 °C), with instant drops of 1 °C between each segment. Additionally, in [24], a three-stage stepped temperature profile was implemented ($T_1$ = 38 °C for 12 h and an immediate drop to $T_2$ = 33 °C for 12 h, followed by an immediate jump to $T_3$ = 48 °C for 24 h) and improved LA production by 1.2 times (using *Escherichia coli* with glycerol as the substrate) compared to conventional constant temperature conditions at 42 °C.

Stepped temperature profiles are difficult to implement in real conditions since instantaneous increases and decreases in temperature are physically impossible to implement in reality. Therefore, the optimal conditions proposed in [23,24] might not be faithfully reproducible in practice. Conversely, smooth and continuous profiles are more suitable for reproduction in a real system. Furthermore, abrupt changes in the physicochemical parameters (like temperature and pH, among others) can negatively affect microbiological systems, leading to cellular stress.

## 4. Conclusions

The Logistic and FOPDT models were adjusted to the growth of *R. oryzae*, and the L-LPwDT and FOPDT-LPwDT models were adjusted to the production of LA at 50% RH with constant temperatures of 22, 35, and 40 °C. In all cases, the hybrid parametric

identification algorithm was applied, obtaining refined results for the kinetic parameters. After identifying the kinetic parameters, the FOPDT model obtained higher $R^2$ values than the Logistic model, so it is a model that can be used in this type of bioprocess. In addition, the parameters $t_p$ and $t_0$ were very useful when correlating the kinetics of fungal biomass growth and LA production, since the concept of a time delay between both kinetics ($t_d$) could be implemented. Consequently, the polynomial relationship of the model parameters ($X_{max}$, $\mu_{max}$, $t_p$, $t_0$, and $Y_{p/x}$) with temperature variation was obtained. This was achieved with a minimum number of trials. This allowed us to establish the mathematical foundations for implementing the dynamic temperature optimization strategy. The variable temperature profile found was smooth, continuous, and differentiable, which is very suitable for SSF bioprocesses, and it increased the LA concentration by 53%. Moreover, its practical implementation is possible because it avoids abrupt temperature changes during the course of the bioprocess. Finally, a very important finding obtained in this work was the change in morphology (from typical mycelial growth to pellet form) presented by *R. oryzae* at 40 °C and 50% RH, a situation that has not been previously reported in other SSF studies. Under these conditions, the $Y_{p/x}$ value improved 1.73 times more than at 35 °C and 0.43 times more than at 22 °C. This further supports the theory that the pellet form is more favorable for increasing LA productivities. This situation would not be expected in a typical SSF and is a very important precedent for the study of fungal morphology changes in SSF and their relationship with LA production.

**Author Contributions:** Conceptualization, M.C.G., S.E.N. and G.S.; methodology, M.C.G. and R.M.G.; software, G.S. and N.P.; formal analysis, M.C.G., S.E.N. and N.P.; investigation, M.C.G.; writing—original draft preparation, M.C.G. and S.E.N.; writing—review and editing, M.C.G., S.E.N., G.S. and N.P.; supervision, S.E.N. and G.S. All authors have read and agreed to the published version of the manuscript.

**Funding:** This article was financially supported by Universidad Católica de Cuyo (UCCuyo; CAM-5-2015; Res. No. 0289-CS-2017) and Secretaría de Ciencia, Tecnología e Innovación (SECITI, SECITI-UCCuyo-2017; Res. No. 0656-CS-2019).

**Institutional Review Board Statement:** This article does not contain studies with human participants or animals performed by any of the authors.

**Informed Consent Statement:** Informed consent was obtained from all individual participants included in the study.

**Data Availability Statement:** All the data are provided in this manuscript.

**Acknowledgments:** The authors are grateful to the staff of the Laboratorio de Control de Calidad Alberto Graffigna of the Universidad Católica de Cuyo for their collaboration and predisposition; they allowed us access to their facilities for the development of different laboratory techniques. The companies SolFrut Alimentos and the winery Tierra del Huarpe S.A. contributed by facilitating the use of some equipment and donated raw material for this research. Likewise, the authors thank the Consejo Nacional de Investigaciones Científicas y Técnicas de Argentina (CONICET), the Universidad Católica de Cuyo, the Universidad Nacional de San Juan, and the Secretaría de Ciencia, Tecnología e Innovación (SECITI, San Juan) for the financial support provided to the María Carla Groff.

**Conflicts of Interest:** The authors declare no conflicts of interest.

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
