# Peer review of "Dynamic Optimization of Lactic Acid Production from Grape Stalk Solid-State Fermentation with Rhizopus oryzae Applying a Variable Temperature Profile"

_fermentation, doi:10.3390/fermentation10020101_

Round 1

Reviewer 1 Report

Comments and Suggestions for Authors

This study has innovatively adjusted First Order Plus Dead Time models for fungal biomass growth, and Luedeking and Piret with Delay Time model was used for lactic acid production. However, There is controversy over the use of quadratic polynomial equations to fit the relationship curve between solid-state fermentation kinetic parameters and time. My questions and suggestions are as follows.

1.      Why choose these three temperatures of 22 ℃, 35 ℃, and 40 ℃ from the perspective of bacterial growth metabolism and kinetics model establishment?

2.      This study assumes that the relationship between kinetic parameters and temperature is a polynomial relationship, and based on this relationship, fits the fermentation kinetics time relationship curve. The basis is that: â‘  the liquid fermentation of this strain conforms to a polynomial relationship; â‘¡ Polynomials are relatively easy in mathematical calculations. (line 205-207)

However, these two reasons are insufficient, as they are:

(1) There is a significant difference between solid-state fermentation and liquid fermentation, as there is no or almost no free water present in solid-state fermentation, which may lead to completely different kinetic laws from liquid fermentation. Therefore, the kinetic relationship of liquid fermentation cannot be directly applied in solid-state fermentation.

(2) Even if the solid-state fermentation of the strain conforms to a polynomial relationship, the degree of the polynomial cannot be determined. Therefore, the direct use of quadratic polynomial fitting in this study has insufficient basis.

(3) Selecting only three temperature points cannot explain the dynamic process. At least five temperatures should be selected, and it is recommended to choose seven temperature points for the experiment.

(4) Polynomial equations have the advantages of easy solving and intuitive curves in mathematics, but they are difficult to combine with cell growth mechanisms, making it difficult to reflect the microbial mechanisms of dynamic parameters.

Therefore, the authors need to fully explain the polynomial relationship and the rationality of selecting three data points.

3.      Starting from line 364, the author discussed the issues of substrate and mycelial morphology, and a new section should be started.

4.      In line 425, the authors mentioned the morphological engineering of the matrix. In this study, the kinetic parameters of strains should also be closely related to substrate characteristics. Can the author briefly explain the influence of substrate characteristics on kinetic parameters?

5.      What’s the basis and mechanism for determining lactic acid using the spectrophotometric technique? I suggest the authors add a corresponding reference in section 2.1.

6.      Why is the detection wavelength 390 nm? What are the advantages and disadvantages of spectrophotometry compared to other LA detection methods such as HPLC and lactate dehydrogenase method?

Comments on the Quality of English Language

The English expression is clear.

Author Response

We thank the referees for the time spent carefully reviewing the manuscript, and for their opinions regarding the science and presentation of the material. 

Please, do not hesitate to contact us for further information about the revised version.

Sincerely yours,

María Carla Groff.

Reviewer 2 Report

Comments and Suggestions for Authors

The manuscript describes the effect of three different temperatures on the kinetic parameters of growth and lactic acid (LA) production by Rhizopus oryzae cultivated under solid state fermentation on grape stalks. Both models have been used by the authors in other published articles (cited in the manuscript) on the production of LA by the same microorganism.

Furthermore, the effect of temperature on the kinetic parameters was evaluated with a polynomial correlation and using this correlation, a dynamic optimization for finding the optimal temperature profile was conducted.

In general the manuscript is well written with logical sequence, supported by results and discussions.

My only comment is that the results in Table 1 should be presented as mean± SD.

Therefore, I will recommend acceptance of the manuscript.

Author Response

(The authors gave the same response as above.)

Round 2

Reviewer 1 Report

Comments and Suggestions for Authors

The manuscript has been well revised, in my opinion, this paper can be published.